

# Cyclone-Induced Surface Ozone and HDO Depletion in the Arctic

Xiaoyi Zhao[1], Dan Weaver[1], Kristof Bognar[1], Gloria Manney[2,3], Luis Millán[4], Xin Yang[5], Edwin Eloranta[6], Matthias Schneider[7], Kimberly Strong[1]

[1]Department of Physics, University of Toronto, Toronto, Ontario, Canada
[2]NorthWest Research Associates, Socorro, New Mexico, USA
[3]Department of Physics, New Mexico Institute of Mining and Technology, Socorro, New Mexico, USA
[4]Jet Propulsion Laboratory, California Institute of Technology, Pasadena, California, USA
[5]British Antarctic Survey, Natural Environment Research Council, Cambridge, UK
[6]Space Science and Engineering Center, University of Wisconsin, Madison, Wisconsin, USA
[7]Institute of Meteorology and Climate Research (IMK-ASF), Karlsruhe Institute of Technology, Karlsruhe, Germany

*Correspondence to*: Xiaoyi Zhao (xizhao@atmosp.physics.utoronto.ca) and Kimberly Strong (strong@atmosp.physics.utoronto.ca)

**Abstract.** Ground-based, satellite, and reanalysis datasets were used to identify two similar cyclone-induced surface ozone depletion events at Eureka, Canada (80.1° N, 86.4° W), in March 2007 and April 2011. These two events were
15 coincident with observations of HDO depletion, indicating that condensation and sublimation occurred during the transport of the ozone-depleted airmasses. Ice clouds (vapour and crystals) and aerosols were detected by lidar and radar when the ozone- and HDO-depleted airmasses arrived over Eureka. For the 2007 event, an ice cloud layer was coincident with an aloft ozone depletion layer at 870 m altitude on 2-3 March, indicating this ice cloud layer contained bromine-enriched blowing snow particles. Over the following three days, a shallow surface ozone depletion event (ODE) was
20 observed at Eureka after the precipitation of bromine-enriched particles onto the local snow pack. A chemistry climate model (UKCA) and a chemical transport model (pTOMCAT) were used to simulate the surface ozone depletion events. Incorporating the latest surface snow salinity data obtained for the Weddell Sea into the models resulted in improved agreement between the modelled and measured BrO concentrations above Eureka. MERRA-2 global reanalysis data and the FLEXPART particle dispersion model were used to study the link between the ozone and HDO depletion. In general,
the modelled ozone and BrO showed good agreement with the ground-based observations, however the modelled BrO and ozone in the near surface layer are quite sensitive to the snow salinity. HDO depletion observed during these two blowing-snow ODEs was found to be weaker than pure Rayleigh fractionation. This work provides evidence of a blowing-snow sublimation process, which is a key step in producing bromine-enriched sea-salt aerosol.

## 1 Introduction

Surface ozone depletion events (ODEs) have been observed in the Arctic spring since the 1980s (Bottenheim et al., 1986; Barrie et al., 1988), and have been linked to halogen chemistry (e.g., Simpson et al., 2007; Abbatt et al., 2012). The halogen chemical chain reactions (R1 to R4) are usually referred to as "the bromine explosion", to indicate the rapidity of the reaction cycle. The speedup of the cycle is due to the conversion of inactive bromide ($Br^-$) to gas-phase reactive bromine (Br), as the net reaction (R5) shows.

$$HOBr + Br^-_{aq} + H^+_{aq} \xrightarrow{mp} Br_2 + H_2O \tag{R1}$$

$$Br_2 \xrightarrow{h\nu} 2Br \tag{R2}$$

$$Br + O_3 \rightarrow BrO + O_2 \tag{R3}$$

$$BrO + HO_2 \rightarrow HOBr + O_2 \tag{R4}$$

$$Net: Br + Br^-_{aq} + H^+_{aq} + O_3 + HO_2 \xrightarrow{mp,h\nu} 2Br + 2O_2 + H_2O \quad . \tag{R5}$$





Here, subscript aq denotes aqueous phase, and mp indicates a multiphase reaction. The activated halogenated species in ODEs can remove gas-phase mercury and deposit the product on snow (Calvert and Lindberg, 2004a, b; Steffen et al., 2008; Toyota et al., 2014; Calvert et al., 2015). The deposited mercury can ultimately reach deep sediments and/or bioaccumulate into wildlife and eventually into humans (Simpson et al., 2007). However, the initial release mechanisms

of the reactive bromine into the atmosphere vary. The facilitating meteorological conditions for each ODE may be very different (Abbatt et al., 2012); low winds with a stable shallow boundary layer (Wagner et al., 2001; Frieß et al., 2004; Lehrer et al., 2004) and high winds with an unstable boundary layer (Yang et al., 2008; Jones et al., 2009; Frieß et al., 2011; Blechschmidt et al., 2016; Zhao et al., 2016a) have both been proposed and observed. These different meteorological conditions can lead to different transport pathways for the low-ozone airmass, thereby changing the

oxidative capacity of the Arctic troposphere. In blowing snow (wind speed > 12 m s$^{-1}$), salty snow aloft can produce sea-salt aerosol (SSA) due to sublimation (Legrand et al., 2016). The bromine-enriched SSA is thought to act as both source and substrate for bromine chemistry (Yang et al., 2008; Yang et al., 2010; Abbatt et al., 2012). In addition, May et al. (2016) report that sea-salt mass concentrations increased in the presence of nearby leads (sea-ice fractures that expose open water) and wind speeds greater than 4 m s$^{-1}$. SSA produced from leads has the potential to alter cloud formation

(May et al., 2016), as well as the chemical composition of the tundra snowpack (Simpson et al., 2005; Pratt et al., 2013). Given the thinning of Arctic sea ice and decreasing of multi-year sea ice extent (Cavalieri and Parkinson, 2012; Stroeve et al., 2012), wind-driven production of SSA from blowing snow (on young sea ice) and/or leads could further increase the frequency of surface ODEs in polar spring.

To understand the high-wind ODE mechanism, measurements of SSA are important. However, direct observation of the

aerosol in blowing snow is challenging due to the meteorological conditions (strong winds and low visibility). Lidar measurements provide a means of looking through the ice clouds and aerosol layer and can provide particle information, for example, size, density, and vertical profile. In addition to aerosol measurements, isotopologue measurements can be a useful proxy for identifying the lifetime of the airmass and aerosol. Measurements of atmospheric hydrogen-deuterium oxide (HDO), which is heavier than H$_2$O, contain information about the history of the air mass. Simultaneous

observations of different tropospheric water isotopologues can provide information on processes related to moisture uptake, exchange, cloud formation, transport/mixing, and temperature upwind of the detected airmass (Schneider et al., 2016). An airmass from the open ocean surface typically contains more HDO than an airmass from the frozen ice-covered ocean at high latitude. For example, in Eureka, from February to April, airmasses with low HDO content are usually from the ice-coved Arctic Ocean, whereas airmasses with high HDO content are usually from ice-free regions of the Atlantic

Ocean. Thus the transport history of the airmass can be evaluated by its water isotopologue ratio. In general, the water isotopologue ratios are expressed in the δ notation, which relates the observed ratio to the standard ratio of ocean water. The HDO/H$_2$O ratio is normally expressed as the relative deviation (δD) from the HDO content of the Vienna Standard Mean Ocean Water (VSMOW) (Craig, 1961), as shown in Eq. (1)

$$\delta D = \left( \frac{R_m}{R_{VSMOW}} - 1 \right) \times 1000\text{‰} ,$$  (1)

where $R$ is the HDO/H$_2$O ratio, subscript $m$ denotes the measurement, and $R_{VSMOW}$ (0.00031152, Craig, 1961) is the HDO/H$_2$O ratio in Vienna Standard Mean Ocean Water.

In the polar regions, ice cloud formation and precipitation can change the HDO/H$_2$O ratio. As with the formation of liquid clouds, when ice clouds form, HDO condenses into the ice more efficiently than H$_2$O, and the air becomes HDO depleted. Thus the scavenging of HDO (removing of the ice condensate) leaves the air depleted in HDO (low δD values) and

provides information about the ice cloud formation (Rayleigh process, Rayleigh and Ramsay, 1894). In addition, blowing snow events can contribute to atmospheric δD change.





During blowing snow events, large numbers of snow particles may get into the air. If the air is not saturated, water can sublime from the particles (depending on wind speed, temperature, relative humidity, snow type, etc., Déry and Yau, 1999, 2001), which reduces their size and eventually leads to the formation of SSA. Yang et al. (2008) parameterised the production flux of SSA, through this process, as a function of the amount of sublimation, snow salinity, snow age, and a

5 few other factors. The sublimation process is normally considered as a non-fractionation process for the ice particles (with no δD changes in the snow/ice crystals). This is because of a low coefficient of self-diffusion of water molecules in ice. When sublimation from ice/snow crystals happens layer by layer, the sublimated molecules have the same δD as when they were frozen in the snow/ice crystals. However, sublimation of blowing snow can change the atmospheric δD through mixing. For example, in Eureka, the mean precipitation δD in March is -265‰ (2005-2013, data provided by F. A.

Michel and X. Feng, personal communication), while the mean atmospheric δD is -442‰ (2006-2014, see Section 3.1.1). Thus, mixing of the sublimed HDO-enriched water vapour (-265‰) from blowing-snow particles with tropospheric background water vapour (-442‰) can increase HDO content, thus contributing to change in the atmospheric δD; and high δD (compared to the Rayleigh fractioning model, see Section 2.1.2) should be expected when observing blowing-snow SSA. This process is similar to the phenomenon that δD of water vapour entering the tropical tropopause layer

(TTL) is higher than that suggested by Rayleigh models (Moyer et al., 1996; Smith et al., 2006; Blossey et al., 2010). The enrichment of HDO in the TTL is associated with deep convection and sublimation of δD-enriched ice from the lower troposphere (Blossey et al., 2010). In summary, the water isotopologue ratio can be used to study SSA formation and transport, and to facilitate the study of blowing-snow ODEs.

Previous case studies identified an ODE initiated from a cyclone in the Beaufort Sea and transported to Eureka on 4-7

April 2011 (Blechschmidt et al., 2016; Zhao et al., 2016a). The meteorological conditions observed at the Eureka Weather Station (EWS) showed that the front arrived at Eureka on 3-4 April. Our previous study showed that the boundary layer on 3-4 April was not stable, and the bromine plume extended from the surface to 2 km. In the present work, by using a nine-year dataset of total column ozone (TCO), column-integrated δD, and the tropopause height, additional features are seen in the 2011 case. The δD measurements show that HDO was also depleted during the April 2011 ODE. A similar

case was identified in the March 2007 data, showing both ozone and HDO depletion. Satellite images and back-trajectories indicate that both cases are linked to a strong tropospheric cyclone.

This work addresses the following scientific questions, using a variety of measurements and atmospheric models:

- Do current blowing-snow ODE modelling results agree with ground-based measurements at Eureka? How well can models simulate BrO, ozone, and sea-salt aerosol compared to measurements?

- The lifetime of reactive bromine is only a few hours in the absence of recycling. Is there evidence of this recycling over aerosol or blowing-snow/ice particles at Eureka?

- The blowing-snow sublimation process is a key step in producing bromine-enriched sea-salt aerosol. Can isotopologue measurements at Eureka be used to provide evidence of this sublimation process?

These questions are important for our understanding of reactive bromine sources, and for meaningful

prediction/modelling of future halogen activation, boundary-layer ozone depletion, and mercury deposition. In the context of a rapidly changing Arctic, these questions are also relevant to environmental change and human health.

**2 Datasets and models**

In this study, ground-based, satellite, and model datasets were combined to produce a comprehensive picture of two ODEs. Ground-based UV and infrared measurements were used to provide ozone column and δD, respectively, above

40 Eureka. Ozone profiles measured by ozonesondes were used to identify surface ODEs. Lidar and radar data were used to





identify the ice cloud and sea-salt particles. Satellite images from the Moderate Resolution Imaging Spectroradiometer (MODIS) were used to identify the location of the cyclone.

### 2.1 Measurements

#### 2.1.1 Ozone measurements

Brewer spectrophotometers measure TCO using direct sunlight at four UV wavelengths (Kerr et al., 1981; Fioletov et al., 2005). The Brewer instrument is the World Meteorological Organization (WMO) Global Atmosphere Watch (GAW) standard for total column ozone measurement (Kerr et al., 1981; 1988). Four Brewers have been deployed at the EWS by Environment and Climate Change Canada (ECCC). Brewer no. 69 is an MKV single spectrophotometer (Adams et al., 2012b) located on the roof of the EWS main building. It was installed in 2004 and has recorded the longest Brewer

dataset at Eureka. The current work uses TCO data from Brewer no. 69 analyzed using the standard Brewer network operational algorithm (Kerr, 2002), with small changes to the analysis parameters due to the high latitude of the site. Normally, high-quality Brewer TCO data need an ozone airmass factor (AMF) less than 3 (Fioletov et al., 2000; Zhao et al., 2016b). However, to cope with the low sun condition in the high-latitude spring, following Adams et al. (2012b), the ozone AMF threshold was increased to 5, which is acceptable under low ozone conditions and allows for more days with

good data in the early spring. The uncertainty in Brewer TCO measurements is typically less than 1% (Fioletov et al., 2005), and for high-quality data (eg., AMF < 3) less than 0.6% (Zhao et al., 2016b).

The EWS has records of electrochemical concentration cell (ECC) ozonesonde measurements since 1992 (Tarasick et al., 2016). Ozonesondes are launched from the EWS weekly year-round and daily during the intensive phase of the Canadian Arctic ACE/OSIRIS Validation Campaigns (2004-2016) (Kerzenmacher et al., 2005; Adams et al., 2012b). The well-

calibrated ECC ozonesondes have precision of 3-5% for total column and ~5-6% uncertainty for the 0-4 km ozone profile (Tarasick et al., 2016). The Brewer and ozonesonde datasets from 2006 to 2014 are used in this study.

#### 2.1.2 Water vapour and HDO measurements

The CANDAC Bruker IFS 125HR is a Fourier transform infrared spectrometer (FTIR) that is part of the Network for the Detection of Atmospheric Composition Change (NDACC, www.ndacc.org). The Bruker 125HR was deployed at the

Polar Environmental Atmosphere Research Laboratory (PEARL) Ridge Lab (86.4 °W, 80.1 °N, Fogal et al., 2013) in 2006 (Batchelor et al., 2009). The PEARL Ridge Lab is 610 m above sea level, 15 km away from EWS (which is 10 m above sea level).

The Bruker 125HR provides measurements of multiple trace gas and water vapour (including $H_2O$ and isotopologues $H_2^{16}O$, $H_2^{18}O$, and $HD^{16}O$, Barthlott et al., 2017; Weaver et al., 2017, in review). The tropospheric column-integrated $H_2O$

and HDO used in the current study were generated as part of the MUSICA (MUlti-platform remote Sensing of Isotopologues for investigating the Cycle of Atmospheric water) project (Schneider et al., 2016) and were used to calculate the δD values (see Eq. 1). Schneider et al. (2016) reported the accuracy of MUSICA δD product to be about 10‰. The FTIR data from 2006 to 2014 are used in the present study.

To further interpret the paired $H_2O$-δD observations, we adapted a method developed by Noone (2012), which uses the

change in δD relative to change in water vapour mixing ratio to provide information about condensation, precipitation, and mixing process in the atmosphere. In general, the method is based on Rayleigh isotope fractioning model (Rayleigh and Ramsay, 1894; Jouzel and Merlivat, 1984)

$$\frac{d\delta}{1+\delta} = (\alpha - 1)\frac{dq}{q} \ ,$$

(2)





where $\alpha$ is the equilibrium isotopic fractionation coefficient, which depends on the temperature of the air parcel (Merlivat and Nief, 1967), and $q$ is the water vapour mixing ratio of the airmass. The Rayleigh model can be applied to both liquid and solid phase formation. The basic assumption of the Rayleigh model is that the condensed phase is formed at isotopic equilibrium with the surrounding vapour and is immediately removed from the airmass after its formation. This is because

droplets can be isotopically modified when evaporation occurs (Ehhalt et al., 1963; Stewart, 1975; Jouzel and Merlivat, 1984; Rozanski et al., 1992; Noone, 2012). The Rayleigh model also assumes that during the formation of ice crystals, there is neither isotopic exchange with the surrounding vapour nor isotopic modification of the ice crystals during their subsequent fallout to the ground (Jouzel and Merlivat, 1984). The additional kinetic fractionation effect (Jouzel and Merlivat, 1984) of ice clouds formed in supersaturation conditions is not applicable for the present study (see Sect. 3.1.2).

MUSICA water vapour isotopologue remote sensing data have two data types (Barthlott et al., 2017). The type-1 data is best suited for tropospheric water vapour distribution studies that disregard isotopologues (e.g., comparison with radiosonde data, analyses of water vapour variability and trends, etc.). The type-2 data is designed for analysing moisture pathways by means of $H_2O$-$\delta D$ pair distributions. Thus the Bruker 125HR/MUSICA water vapour isotopologue type-2 data from 2006 to 2014 were used in the present study. Similar to the Brewer TCO dataset, the SZA filter of the water

vapour isotopologue dataset has also been relaxed to expand the temporal coverage of dataset for these high Arctic conditions.

### 2.1.3 Cloud and aerosol measurements

MODIS on board the NASA Terra-1 satellite measures visible and thermal electromagnetic radiation. In this work, false colour images constructed using bands 2, 3, and 31 are used to identify cyclonic polar low systems

(http://modis.gsfc.nasa.gov). The Arctic High Spectral Resolution Lidar (AHSRL) and Millimetre-wave Cloud Radar (MMCR) measurements were used to provide cloud and aerosol information. The AHSRL and MMCR were located at the Zero Altitude PEARL Auxilary Laboratory (0PAL), which is located at the northwest corner of the EWS site. The AHSRL was developed at the University of Wisconsin and deployed at Eureka from August 2005 to 2010. It has a frequency-doubled diode-pumped Nd:YAG laser at 532 nm (Bourdages et al., 2009), and measures the particle

backscatter cross-section ($\beta_{lidar}$) and circular depolarization ratio, which can be used to differentiate between spherical liquid droplets and crystalline particles. The MMCR (Shupe et al., 2010) has been deployed at Eureka since 2005. MMCR measures Doppler velocity, Doppler spectra, spectral width, and equivalent radar reflectivity for clouds, from which the particle backscatter cross-section ($\beta_{radar}$) can be determined (Bourdages et al., 2009). Bourdages et al. (2009) proposed a method to categorise atmospheric particles and their mixtures by combining information from the lidar and radar

measurements. They calculated a colour ratio defined as the ratio between the radar and lidar backscatter cross-sections

$$R_{colour} = \frac{\beta_{radar}}{\beta_{lidar}} \quad . \tag{3}$$

The colour ratio is an average property for particles in a measurement volume, and is a good proxy for particle size (Bourdages et al., 2009). The MMCR measures reflectivity from 90 m to 20 km in altitude, and is sensitive to volume backscatter cross-sections greater than $10^{-14}$ $m^{-1}$ $sr^{-1}$. Following Bourdages et al. (2009), an interpretation in terms of

particle size is possible for the colour ratio in the range of $10^{-9}$ to $10^{-3}$, which corresponds to a particle size of about 10 to 150 μm. This colour ratio method cannot provide particle size information for fine and medium aerosol particles (radius less than 10 μm), but these aerosol layers can be distinguished by $\beta_{lidar}$ smaller than $2\times10^{-5}$ $m^{-1}$ $sr^{-1}$. In the present work, we adapted this colour ratio method to distinguish between aerosols, ice clouds, and ice crystals.



## 2.2 Models

### 2.2.1 MERRA-2

The second Modern-Era Retrospective analysis for Research and Applications (MERRA-2) is an atmospheric reanalysis from NASA's Global Modeling and Assimilation Office (GMAO) that provides high-resolution globally gridded meteorological fields using the Goddard Earth Observing System-Version 5 data assimilation system (e.g., Bosilovich et al., 2015; Fujiwara et al., 2017). MERRA-2 has a horizontal resolution of $0.625°$ in longitude and $0.5°$ in latitude. In the present work, vertical profiles of MERRA-2 ozone, temperature, pressure, and scaled potential vorticity (sPV, potential vorticity scaled in "vorticity units" to give a similar range of values at each level, e.g., Dunkerton and Delisi, 1986; Manney et al., 1994; Adams et al., 2013) over Eureka were derived from the MERRA-2 assimilated state variable (ASM) data collection (GMAO, 2015). The profile data are on 72 model layers with approximately 1-km vertical spacing near the tropopause, and 3-hour temporal resolution. Tropopause locations are calculated from MERRA-2 according to dynamical (the PV-based, so-called dynamical tropopause) and temperature gradient (so-called thermal tropopause) definitions using the JEt and Tropopause Products for Analysis and Characterization (JETPAC) package described by Manney et al., 2011; here we used the thermal tropopause height (WMO, 1992) in Section 3.1.

### 2.2.2 pTOMCAT

The Cambridge Parallelised-Tropospheric Offline Model of Chemistry and Transport (pTOMCAT) is a global 3-D chemistry transport model (CTM) (O'Connor et al., 2005; Yang et al., 2005). The forcing files (temperature, wind, and humidity fields) for pTOMCAT are 6-hourly ERA-Interim data from the European Centre for Medium-Range Weather Forecasts (ECMWF) (Dee et al., 2011; Fujiwara et al., 2017). Monthly sea-ice coverage and sea surface temperatures are taken from the Hadley Centre Sea Ice and Sea Surface Temperature (HadISST) dataset (Rayner et al., 2003). The model's horizontal resolution is $2.8° \times 2.8°$ (longitude $\times$ latitude) with 31 vertical layers from the surface to about 10 hPa at the top layer.

A detailed process-based SSA scheme has been implemented in the model (Levine et al., 2014) based on the work of Reader and McFarlane (2003). Since the Levine et al. (2014) work, some updates have been introduced to the model, including improved precipitation, dry deposition velocities on snow for inorganic bromine species, and reduced snow salinity applied to the blowing snow (Legrand et al., 2016). Both open-ocean sourced and sea-ice sourced SSA (OO-SSA and SI-SSA) are tagged in order to track their history. Both OO-SSA and SI-SSA are in 21 size bins covering the range of 0.1-10 μm in dry radius (Rhodes et al., 2017). To study the transport of the SSA in the present work, SI-SSA with dry radius at 0.25, 1, and 5 μm were used in the comparison with another particle dispersion model.

The tropospheric bromine chemistry scheme in pTOMCAT is based on the work of Yang et al. (2005; 2010). The bromine source includes inorganic sea-salt from both the open ocean and the sea-ice zone (from blowing snow, Yang et al., 2008), and halocarbons from long-lived (e.g. $CH_3Br$) and very short-lived substances (VSLSs). The emissions of the the VSLSs ($CHBr_3$, $CH_2Br_2$, $CH_2BrCl$, $CHBr_2Cl$, and $CHBrCl_2$) are based on the original work (scenario 5) of Warwick et al. (2006), except for emissions of $CH_2Br_2$, which were updated to 50% of the original flux to allow a closer match to the observation (Yang et al., 2014). Heterogeneous reactions that reactivate inactive inorganic bromine species (such as HBr) to active Br radicals, on both atmospheric background particles (offline data taken from Heintzenberg et al., 2000) and on the SSA produced from sea spray and blowing snow (online calculated) have been included in this model integration. In general, the bromine flux from SSA depends on: 1) the rates of SSA production from both open ocean and blowing snow on sea ice, and 2) the bromine depletion factors for SSA, which are size dependent as observed (Yang et al., 2008; Yang et al., 2010). The SSA production rate is calculated using the rate at which snow particles are lifted from





the surface and subsequent sublimation (which depends on wind speed, temperature, relative humidity, and the age of snow). More details about the calculation of bromine release flux from blowing snow can be found in Zhao et al. (2016a). The current work uses the transported SI-SSA information from pTOMCAT to make a direct comparison with ground-based measurements.

### 2.2.3 The UKCA model

UKCA is a global chemistry-climate model (CCM); its dynamical core is the Met Office Unified Model (UM) version 7.3 running in the HadGEM3-A configuration (Morgenstern et al., 2009). The UMUKCA-CheST version contains a comprehensive stratospheric chemistry scheme as well as a detailed tropospheric chemistry scheme, including isoprene chemistry and improved bromine chemistry in both the troposphere and stratosphere (Braesicke et al., 2013; Yang et al., 2014). The model's horizontal resolution is $3.75\,° \times 2.5\,°$ with 60 vertical layers from the surface to about 84 km. A nudged UKCA version (forcing by ERA-Interim, the same meteorological fields as pTOMCAT) is used in this study. Tropospheric bromine chemistry was introduced to UKCA based on the work in pTOMCAT (Yang et al., 2005; Yang et al., 2010). In general, UKCA and pTOMCAT share the same SSA production flux from the same particle size bins (and thus have same bromine release flux). However, unlike pTOMCAT, UKCA does not trace the SSA transport after the emission, therefore there is no online SSA being taken into account in the heterogeneous rate calculation (only a monthly climatology aerosol dataset from the CLASSIC aerosol scheme was used, Johnson et al., 2010). In terms of dynamics, the UM model is much different to the atmosphere-only pTOMCAT in many aspects, for example, from boundary layer vertical mixing scheme to the cloud parameterisation, see more details in Russo et al. (2011) and Ruti et al. (2011).

The UKCA model successfully simulated the 2011 blowing-snow-induced ODE for our previous study (Zhao et al., 2016a). The difference between the present work and our previous study (Zhao et al., 2016a) is mainly snow salinity, as this parameter has recently been updated based on samples collected in the Weddell Sea (Legrand et al., 2016). The new surface snow salinity (top 10 cm snow layer, about 0.3 practical salinity unit, the same value as pTOMCAT) in the present work is smaller by an order of the value used in Zhao et al. (2016a), which was the column mean salinity (Sander et al., 2003; Yang et al., 2008).

### 2.2.4 FLEXPART

The Lagrangian FLEXible PARTicle dispersion model (FLEXPART) was first released in 1998 for calculating the long-range and mesoscale dispersion of air pollutants from point sources. It has been used to examine source regions for aircraft, satellite, ground-based station, and ship-based studies (e.g., Stohl et al., 2005; Begoin et al., 2010; Gilman et al., 2010; Blechschmidt et al., 2016; Lutsch et al., 2016). In the present work, FLEXPART is run backwards in time for a group of passive SSA tracers that are transported by winds from $0.5^o \times 0.5^o$ resolution NCEP Climate Forecast System Version 2 (CFSv2) 6-hourly products (http://rda.ucar.edu/datasets/ds094.0/, Saha et al., 2014). To compare with pTOMCAT modelled SSA, we used three different-sized SSA tracers (0.25, 1, and 5 µm) in the simulation, and released them at Eureka on two different grids. One grid was $0.4^o \times 0.4^o$ (~20 km $\times$ 20 km) and had Eureka at the centre of the grid, and the other grid $2.8^o \times 2.8^o$ (~56 km $\times$ 310 km) was selected to match one of the pTOMCAT grids that includes Eureka.



## 3 Results

### 3.1 Surface ozone depletion and water vapour measurements

In the atmosphere, most ozone (~90%) is in the stratosphere. Thus the TCO is a function of tropopause height, with a low tropopause generally increasing TCO. In contrast, $H_2O$ and HDO are most abundant in the lower troposphere, thus

precipitable water vapour (PWV) and column-integrated $\delta D$ may provide information about the meteorological conditions in the lower troposphere. TCO, thermal tropopause height, and $\delta D$ can be combined in a 3-D scatter plot that contains dynamical and chemical information. Figure 1a shows the 3-D scatter plot generated by merging nine years of Brewer TCO, MERRA-2 tropopause height, and Bruker 125HR $\delta D$. The data points are colour coded by PWV amount (measured by the Bruker 125HR). The 3-D scatter plot reveals the general linear relation between each pair of the three variables as

discussed above. The low $\delta D$ outliers marked on Fig. 1a are on 5 and 6 April 2011 (when $\delta D$ values are -503‰ and -505‰ respectively), suggesting unusual meteorological conditions (cyclone events).

The size of the combined dataset is limited by the Brewer TCO measurements, which only start in late March. To expand the dataset, the MERRA-2 TCO was used instead of Brewer TCO. The MERRA-2 TCO for Eureka has a high correlation (R = 0.99) and a low positive bias (1.6%) compared to Brewer TCO (not shown here). The resulting increase in the

number of coincident data points is seen in Fig. 1b, which reveals a similar event for 3 and 4 March 2007 (-529‰ and -551‰ for $\delta D$ values respectively). Figure 2 shows the whisker plot of Bruker 125HR monthly $\delta D$ and PWV measurements. The median (mean) $\delta D$ values for March and April at Eureka are -443 (-442) and -410 (-408)‰ respectively. The $\delta D$ value describes the relative deviation of HDO content from the standard mean ocean water. Thus the HDO value on 4 March 2007 is 55.1% less than the standard; the $\delta D$ value is 24.7% less than the March mean

(comparing -551‰ and -442‰). Similarly, the $\delta D$ value on 5 April 2011 is 23.8% less than the monthly mean of April (comparing -503‰ and -408‰). The coincidence of these two events (2007 and 2011) on the 3-D plot (Fig. 1b) indicates they may share similar dynamical and chemical causes. Figure 2a shows that the median (mean) PWV is 1.4 (1.7) mm in March and 2.5 (2.6) mm in April. The daily mean PWV value was only 0.5 mm on 4 March 2007 and was 1.0 mm on 5 April 2011, which indicates relatively dry conditions for these two events. More detailed discussion of HDO depletion is

presented in Section 3.3.

Figure 3(a and b) shows Eureka ozonesonde records from February to March 2007 and March to May 2011. The 0-4 km ozone profiles on 3 and 4 March 2007 and 4 and 6 April 2011 are indicated in the figure. Surface ozone depletion during these two events can be seen, although the 2007 event is weaker than the 2011 event. Figure 3(c and d) shows 0-8 km relative humidity (RH) values measured by radiosondes. The low RH and PWV during these two events (see details in

Section 3.3.1) can facilitate SSA production though the blowing-snow mechanism proposed by Yang et al., (2008).

### 3.1.1 Chemical model results

Figure 4 shows the ozone and BrO volume mixing ratio profiles over Eureka for the 2007 event from ozonesondes and model simulations. A shallow surface ozone depletion layer (see the UKCA result in Fig. 4b, minimum value ~5 ppbv) can be found similar to the ozonesonde measurements (see Fig. 4a, minimum value ~1 ppbv) from 3 to 5 March. The

ozonesondes measured 40-50 ppbv background ozone for 2-4 km, but both the pTOMCAT and UKCA modelled about 10 ppbv less at these altitudes. The BrO concentrations from both models are less than 5 pptv. One interesting feature in Figure 4 is that ozone depletion was also found from 1 to 2 March. The ozone depletion layer on 1 and 2 March is aloft above the surface, with a maximum depletion at ~800-900 m (see Figure 5). However, this aloft layer is not captured by either model.



Figure 6 shows the ozone and BrO volume mixing ratio profiles over Eureka for the 2011 event. The highlight of the re-run of the UKCA model for 2011 is the increased BrO volume mixing ratio. In our previous study (Zhao et al., 2016a), UKCA only modelled about half of the maximum BrO VMR compared to ground-based measurements. The previous sensitivity study found that even doubling the bromine release flux from the source region in the model did not increase

the BrO VMR over Eureka because of very low ozone VMR (< 1ppbv) in this case (Zhao et al., 2016a). However, in this re-run, the decreased snow salinity in the model input (which decreases the bromine release flux from the source region) surprisingly increased the modelled BrO VMR (to ~23 pptv) over Eureka, which is comparable to the ground-based MAX-DOAS measurements of ~21 pptv (the UKCA previously modelled ~13 pptv, see Zhao et al., 2016a for more details). The reduced bromine flux from the source region reduced the depletion of ozone along the trajectory, which in

turn preserved the high concentration of bromine. In general, this result supports the hypothesis of Zhao et al. (2016a) that during a strong surface ODE (ozone VMR < 1 ppbv), the bromine concentration and partitioning can be significantly affected by the near-zero ozone concentration. pTOMCAT also captured the 2011 event, but the results are not as good as those of the UKCA model. This could be due to the difference in the models' dynamics in the boundary layer (Hoyle et al., 2011; Ruti et al., 2011), model resolution, and the background aerosol scheme (Yang et al., 2005; Yang et al., 2010;

Zhao et al., 2016a).

Comparing the measured and modelled ozone profiles for the two events (see Figs. 4 and 6), the surface ozone depletion on March 2007 is weaker than that during the April 2011 event. The weaker ODE in 2007 is likely due to insufficient sunlight. The first sunrise at Eureka is on 21 February. On 4 March, the daylight length is only 7.5 hours and the maximum solar elevation angle is only 3.7°, but on 4 April the daylight length is 17.5 hours and maximum solar elevation

angle is 16°. Note that the insufficient sunlight will make reaction R2 the rate-limiting step in the bromine reaction cycle (R1 to R4). In addition, the strength of the two cyclonic polar low systems was different (discussed in Section 3.1.3).

### 3.1.2 Airmass history

To examine the history of the ozone-depleted airmass and aerosol transport, FLEXPART (Stohl et al., 2011) is used in the present study. Figure 7 shows some of the FLEXPART SSA-TRACER total column sensitivities, which, when multiplied

with emission flux, provides a simulated concentration at the receptor (Stohl et al., 2013). As knowledge of the SSA type is limited, simulations are kept as simple as possible. The SSA-TRACER had a scavenging scheme (both wet and dry deposition) applied (Seibert and Frank, 2004; Stohl et al., 2005; Stohl et al., 2011), but had no assumption on its lifetime. We used 20,000 tracer particles for each release. The release times were set to be coincident with the Bruker δD measurements, and the duration of each release was one hour. Each backwards run lasts for 6 days, and the release

heights were set to 0 to 500 m and 1500 to 2000 m respectively. The particle release location areas were set to use the coarse grid from pTOMCAT (which includes Eureka) and a much finer grid (only 2% of pTOMCAT grid size, not shown here) as described in Sect. 2.2.4. The aerosol tracers released in all backwards simulations show sensitivity to the Beaufort Sea region, and the plumes reveal the structure of the cyclones (see Fig. 7).

MODIS false colour images (Fig. 8) show cloud tops of two polar cyclones, which formed over the Beaufort Sea during

both periods of interest. The ERA-Interim data (see Fig. 9) show that the wind speed in the cyclone increased to 20 and 24 m s⁻¹ for the 2007 and 2011 cases respectively; and the boundary layer heights increased to 710 and 800 m for those two events, respectively. Thus the cyclone in 2007 was weaker than the one in 2011. In general, these observations and model simulations confirm that the ozone-depleted airmass and aerosols that were observed at Eureka during those two events were generated in high wind conditions in the Beaufort Sea region and then transported by the polar cyclones.





### 3.2 Cloud and aerosol

Among many different proposed bromine sources (Abbatt et al., 2012), we focused on the SSA in this study. In the Arctic spring, the SSA concentration depends on meteorological, sea-ice, and snow conditions (Nilsson et al., 2001; Rannik, 2001; Lewis and Schwartz, 2004; Yang et al., 2008; May et al., 2016). Wind speed, RH, sea surface and air temperature

gradient, water/ice/snow salinity, and removal processes (wet and dry deposition) will all affect the SSA formation and concentration. For the present work, the meteorological conditions for the 2007 and 2011 events support the work of Yang et al. (2008), which reported that SSA production rate from snow can be significant in high wind conditions.

### 3.2.1 Lidar and radar observations

The lifetime of SSA in the atmospheric boundary layer can range from minutes to days, depending on its size and

meteorological conditions. Fine SSA (diameter < 2.5 μm) has slower deposition rates compared to medium-sized SSA (diameter from 2.5 to 10 μm). Following the work of Yang et al. (2008), large SSA (diameter > 10 μm) was not included in pTOMCAT and UKCA due to their short lifetime (faster dry deposition rate) in the atmosphere. In the Arctic spring, the low temperature makes the SSA contributed from the open ocean less important. Thus although pTOMCAT includes both OO-SSA and SI-SSA, only the latter is considered in this study. For these two events, both ground-based

observations and model data revealed high aerosol concentration during the ODEs. For the 2011 case, the aerosol extinction measurements from a ground-based MAX-DOAS instrument and the cloud measurements from the MMCR were presented in our previous study (Zhao et al., 2016a). In the present work, we compare the pTOMCAT SI-SSA data during the 2007 event with aerosol and cloud measurements from the AHSRL and MMCR (note that the MAX-DOAS measurements started in 2010, and the AHSRL measurements ended in 2010).

Figure 10 shows the comparison of lidar/radar profile data with the pTOMCAT SI-SSA VMR profile. Figure 10a shows the measurement time for the ozonesondes (dashed lines) and Bruker 125HR (green lines/boxes). Figure 10b and c show the linear depolarization and backscatter cross-sections measured by the lidar. The backscatter cross-sections measured by the lidar ($\beta_{lidar}$) and radar ($\beta_{radar}$) were used to calculate the colour ratio (see Figure 10d).

The cyclonic airmass arrived at Eureka on 18:00 UTC 1 March. AHSRL linear depolarization data indicate that there was

an aloft layer of particles (Fig. 10b, area α) that showed high depolarization from 30 to 50% at 700-1000 m. The colour ratio (Fig. 10d) for this layer was in the range of $10^{-5}$ to $10^{-3}$, indicating that it was ice cloud (vapour and ice crystals) with ice precipitation (see Fig. 10 in Bourdages et al. (2009) for the classification chart), and that the effective radius of the particles was in the range of 50-150 μm. This layer of ice cloud was coincident with the aloft ODE layer (for example, the ozonesonde at 23:28 on 1 March has the greatest ozone depletion (to 7 ppbv) at 870 m) as shown in Figure 11. Bourdages

et al. (2009) described the lidar and radar measurements from 4-5 March 2007 in detail. An ice crystal layer (high depolarization, low colour ratio) is seen from the surface to 300 m from 14:00 to 24:00 UTC on 4 March (Fig. 10d, area γ). A thin aerosol layer (low depolarization, low colour ratio, 300-500 m) for that period is also seen (Fig. 10c, area β2). Ice clouds were identified in the middle troposphere from 16:00 on 4 March to 8:00 on 5 March UTC (Fig. 10c, area β3). These shallow layers of ice crystal (area γ) and aerosol (area β2) were coincident with the shallow ODE layer (for

example, the ozonesonde at 23:16 on 4 March shows ozone depletion to 1 ppbv from the surface to 200 m, with the ozone mixing ratio increasing to 32 ppbv at 650 m) as shown in Figure 11. Another two aerosol-enhanced events were also observed as indicated in Fig. 10c (areas β1 and β4). In addition, small water clouds were observed at 2.2 to 2.5 km between 10:00 and 14:00 UTC on 5 March and at 2.4 km at about 1:00 UTC on 6 March. A very thin water cloud was observed at about 1.8 km between 8:00 to 14:00 UTC on 6 March.

The lidar/radar observations of the ice clouds, ice crystal, and boundary layer aerosol from the cyclonic airmass explain why strong HDO depletion was observed coincidently with the arrival of the front. Interestingly, the ice cloud was at the





same height as the aloft ODE layer. The lidar/radar observations also reveal deposition (falling snow and ice crystals) processes at Eureka; the ice precipitate has high depolarization and vertically aligned fall streaks. The prolonged surface ODE on 4 and 5 March could be due to the deposition of bromine-enriched particles onto the local snow pack.

### 3.2.2 Model results

The modelled SI-SSA from pTOMCAT in three bins (0.25, 1, and 5 μm) captured some features of the transported aerosols during this period (Fig. 10e to g). For example, the three aerosol events seen in Fig. 10c (areas β1, β2, and β4) can also be found in the modelled SI-SSA. Modelled 5 μm SI-SSA is most abundant in area β1, whereas 1 μm and 0.25 μm SI-SSA are most abundant in area β4. Back-trajectory results show that the fine aerosol on 6 March was transported from the north, whereas the medium size SSA on 1 March came from the south (not shown here). A weak shallow aerosol

layer on 4 March was also captured by pTOMCAT(see area β2), and it was transported from the west (as from the cyclonic airmass).

As discussed in Sect. 3.1, the modelled and observed ozone depletion for the 2007 event is weaker than that for 2011 event. In addition, pTOMCAT modelled ozone depletion was less than that in the UKCA model (see Figure 5). One possible explanation is that pTOMCAT didn't fully capture the production of SSA during the blowing snow. Figures 7

and 9 show that the high-wind areas for the 2011 event were over sea ice, whereas the high-wind areas for the 2007 event were close to land. Because of the coarse model grid resolution, both models underestimated the production of SSA in that close-to-land cyclone case. Thus the underestimated SSA may reduce the strength of the ODEs.

As discussed in Sect. 3.2.1, none of the chemical models used in the present study reproduced the aloft ozone depletion layer. pTOMCAT results show that there was almost no SI-SSA during that period (see Fig. 10e-g, area α). Thus the

production and deposition mechanisms of SI-SSA in the models still need further investigation. For example, the SI-SSA deposition rates (wet and/or dry) might be too high in this case. In general, this comparison of lidar/radar measurements and modelled SI-SSA provides supports for the blowing-snow SSA production mechanism, and suggests that further improvements in SI-SSA modelling are important to improve blowing-snow ODE simulations.

### 3.3 HDO depletion

**3.3.1 δD-PWV observation**

Figure 12 shows the evolution of δD as a function of PWV during the two cyclone events. The grey dots are nine-year δD-PWV Bruker 125HR measurements (all data, 2006-2014). The red and blue dots are daily mean δD-PWV values for the dates indicated. The green dot in Fig. 12a (b) represents March (April) mean δD-PWV values over all nine years. Other Rayleigh curves with different phase equilibrium ice-vapour fraction coefficients ($\alpha_{\text{e-ice-vapour}}$) are indicated by

coloured dashed lines, with Fig. 12a using the δD-PWV daily mean on 1 March 2007 as the origin and Fig. 12b using the δD-PWV daily mean on 2 April 2011 as the origin. The dashed green ($\alpha_{\text{e-ice-vapour}} = 1.198$) and blue ($\alpha_{\text{e-ice-vapour}} = 1.228$) Rayleigh curves correspond to condensation processes at temperatures of -30 and -40 $^{\circ}$C (Merlivat and Nief, 1967). The error bars on the red and blue dots are standard deviations of the δD-PWV values used to calculate the daily mean.

Figure 12a shows an HDO depletion process from 28 February to 4 March 2007 (red dots), followed by a moistening

process from 5 to 6 March (blue dots). The HDO depletion from 2 to 4 March 2007 corresponds to the condensation process in the cyclonic airmass, which was captured by the Rayleigh fractioning model (see the green and blue Rayleigh curves). Tropospheric RH measured by radiosondes is below 75% from 2 to 4 March 2007, and temperatures in the 0-4 km cloud layers were -35 to -45 $^{\circ}$C during the period. Thus kinetic fractionation due to supersaturation in ice clouds is not applicable for this case. The deviation of the δD-PWV pair from Rayleigh curves (green and blue dashed lines) on 3

March 2007 is possibly due to sublimation of HDO-enriched blowing snow. The water vapour sublimation flux from





blowing snow is 0.5 mm/day, with conditions of 10 m/s wind speed, -20 °C ambient temperature, and 80% RH. With conditions of 20 m/s wind speed, -20 °C ambient temperature, and 75% RH, the water vapour sublimation flux from blowing snow can increase to 2.6 mm/day. These amounts of water vapour sublimation flux from δD-enriched blowing snow are strong enough to affect the atmospheric δD value (since the PWV on 4 March 2007 is only 0.5 mm) through

mixing. The temporal mixing line method (Keeling, 1958; Miller and Tans, 2003; Noone et al., 2013) can be used to estimate the δD and/or sublimation flux of blowing snow. However, isotopologue measurements for precipitation at the source region (for example, near the Beaufort Sea) are necessary to validate the results and to make meaningful comparison of measured with modelled sublimation flux. In general, more detailed water isotopologue modelling and observations are necessary to further evaluate the blowing–snow HDO evolution.

Figure 12a also shows a remoistening process that started on 5 March (see blue dots), which was due to the cyclonic airmass starting to mix with a different airmass transported from the north. This mixing process is also coincident with the termination of the ODE. As shown in Figure 11, the shallow surface ODE became weaker on 5 March than that on 3 and 4 March, and terminated on 6 to 7 March, as the surface ozone VMR increased from 11 to 15 ppbv.

Figure 12b shows an HDO depletion process from 1 to 6 April 2011. The depletion process from 2 to 5 April is also

deviated from Rayleigh process (green and blue dashed lines) on 3 April (the measured 0-4 km cloud temperature is -30 to -35 °C). Tropospheric RH measured by radiosondes is below 80% from 3 to 5 April 2011. Surface clouds containing ice crystals were observed for the 2011 case (Zhao et al., 2016a), and other meteorological conditions were also similar to the 2007 case. Thus, as with the 2007 event, the sublimation of HDO-enriched blowing snow is possibly the cause of this measured δD-PWV deviation from Rayleigh process; the measured fractioning result on 3 April 2011 appears to be

weaker (less HDO loss) than modelled green and blue Rayleigh fractioning lines.

In general, the δD-PWV values provide useful information regarding the history of the cyclonic airmass. HDO depletion on 3 March 2007 and 3 April 2011 (during two ODE events) was found to deviate from (weaker than) pure Rayleigh fractionation, suggesting that sublimation likely contributed to the δD evolution.

### 3.3.2 Influence of the polar vortex

The previous sections described surface ODEs coincident with the observation of depleted HDO and increased SSA. Derived Meteorological Products (DMPs), similar to those described by Manney et al. (2007), were calculated using the JETPAC package (Manney et al., 2011) for above Eureka and along the lines-of-sight of the Bruker instrument (Lindenmaier et al., 2012) using MERRA-2 analysis. Figure 13 shows the MERRA-2 temperature profiles over Eureka with the boundaries of the polar vortex indicated (defined by an sPV value of $1.6 \times 10^{-4}$ s$^{-1}$ for the inner edge of the vortex,

Manney et al., 2007). Both the 2007 and 2011 events show strong cooling (210-220 K) in the stratosphere from 15-25 km, which is caused by the presence of the stratospheric polar vortex overhead (Manney et al., 2008; Adams et al., 2013). In both cases, the stratospheric vortex and the tropospheric cyclonic low-pressure system were present at the same time. MERRA-2 captured the surface pressure and temperature changes in the polar lows, as seen from the decreased surface temperature in Fig. 13a for 3-4 March 2007 and in Fig. 13b for 5-6 April 2011. MERRA-2 PV maps (not shown here)

indicate that the edge of the elongated vortex was over Eureka for both years. Cold stratospheric airmasses are more HDO depleted than warm airmasses. Since the extremely low δD values measured during these two events were coincident with the presence of the stratospheric vortex, the δD sensitivity in the stratosphere was examined.

The Bruker 125HR water vapour measurements have limited sensitivity in the stratosphere and therefore the δD data should have limited sensitivity to the presence of the polar vortex. We investigated the δD sensitivity to the airmass being

inside and outside the stratospheric vortex. Following Manney et al. (2007) and Adams et al. (2012a), sPV values along the lines-of-sight of the Bruker instrument were interpolated to the 490-K potential temperature level (lower stratosphere,





~19 km), with the inner vortex edge defined by $sPV_{490K}$ values of $1.6 \times 10^{-4}$. With the $sPV_{490K}$ values, the Bruker measurements can be categorised into two regimes: inside and outside the polar vortex.

Figure 14 shows the whisker plots of Bruker δD and PWV data. Figure 14a and b show the δD and PWV data in March of all nine years. The median δD outside the vortex in March is $-438^{+23}_{-31}$‰ (75th to 25th percentile), while the value inside

the vortex is $-453^{+28}_{-24}$‰. Thus statistically, there is no significant difference of δD in these two regimes. In Figure 14c and d, the period is expanded to all spring measurements (from February to April, by which time the stratospheric vortex has broken up), and these panels also confirmed that the Bruker δD and PWV data are not sensitive to the presence of the polar vortex. Moreover, no evidence was found in the ozonesonde ozone and radiosonde RH measurements to indicate the existence of a stratospheric intrusion, which could cause low HDO. Thus to summarize, this sensitivity test and

observations indicate that the extremely low δD measured during the two events studied here were caused by tropospheric HDO depletion.

**4 Conclusion**

Data from six instruments and four models were used to investigate Arctic ODEs. Using the TCO, tropopause height, and δD data, two similar cyclone events that show both depleted surface ozone and HDO at Eureka were identified from the

15 nine-year dataset: in March 2007 and April 2011. The FLEXPART particle dispersion model was used to simulate airmass transport. The aerosol tracers released in all backwards simulations show sensitivity to the Beaufort Sea region, and the plumes revealed the structure of the cyclones. The ERA-Interim data show that the wind speed in the cyclone increased to 20 and 24 m s$^{-1}$ for the 2007 and 2011 cases respectively, and the boundary layer heights increased to 710 and 800 m respectively.

In general, the ground-based observations show good agreement with modelled ozone and BrO data. One key result is that the modelled BrO and ozone in the near-surface layer are quite sensitive to the snow salinity. Newly incorporated surface snow salinity data increased the modelled BrO concentration above Eureka to agree with our previous measurements for the 2011 event (Zhao et al., 2016a). This is due to the reduced bromine flux from the source region, which, as a result, reduced the depletion of ozone along the trajectory and preserved a higher concentration of bromine

oxide. This result supports the conclusion of that study that during a strong surface ODE (ozone VMR < 1 ppbv), BrO concentration can be significantly controlled by the ozone concentration.

Modelled blowing-snow SSA was compared with lidar/radar measurements showing that pTOMCAT SI-SSA data captured some features of the increased aerosol for the 2007 and 2011 events. For the 2007 event, the lidar and radar observed ice cloud (vapour and ice crystals) from 700 to 1000 m height (from the end of 1 March to 3 March), which is

30 coincident with an aloft ozone depletion layer having the greatest ozone depletion (to 7 ppbv) at 870 m. However, pTOMCAT and UKCA both failed to reproduce this feature in their modelled ozone profiles. The pTOMCAT results show that there was almost no modelled SI-SSA during that period, indicating limitations in the model simulation of aerosol and that further investigation is needed, e.g., into the production and deposition mechanisms of SI-SSA.

The 2007 lidar/radar observations also reveal deposition (falling snow and ice crystals) from this ice cloud at Eureka.

Thus the following shallow surface ODE (0 to 200 m) on 4 and 5 March was likely due to the deposition of bromine-enriched particles onto the local snow pack. The detailed vertical structure of ice clouds, ice crystal, and aerosol layers obtained for the 2007 event could be used in future bromine modelling comparisons.

HDO depletion observed during these two blowing-snow ODEs is found to be weaker than pure Rayleigh fractionation. The evolution of δD-PWV was used to distinguish cyclone-originating airmasses from non-cyclone-originating airmasses,

indicating that the termination of the shallow surface ODE is due to mixing with a different airmass. Although the edge of



the stratospheric polar vortex was found over Eureka during those periods, no evidence was found that the low δD value was caused by the polar vortex or stratospheric intrusions. This work thus provides evidence of a blowing-snow sublimation process, which is a key step in producing bromine-enriched SSA. In general, this work improved our understanding of ODE processes by combining a variety of measurements with atmospheric models, and it facilitated both improved modelling of the atmosphere and the interpretation of the measurements.

**Acknowledgements and Data**

Data from the Eureka instruments are available through Canadian Network for the Detection of Atmospheric Change (CANDAC, http://www.candac.ca/), which has been supported by ARIF, AIF/NSRIT, CFCAS, CFI, CSA, ECCC, GOC-IPY, INAC, NSERC, NSTP, OIT, ORF, PCSP, and SEARCH. AHSRL and MMCR data are available from http://hsrl.ssec.wisc.edu/. Any additional data may be obtained from Xiaoyi Zhao (xizhao@atmosp.physics.utoronto.ca). X. Zhao was supported by the NSERC CREATE Training Program in Arctic Atmospheric Science and the Probing Atmosphere in the High Arctic (PAHA) program. The MUSICA NDACC/FTIR retrievals have been made in the context of the project MUSICA, which was funded by the European Research Council under the European Community's Seventh Framework Programme (FP7/2007-2013) / ERC Grant agreement n° 256961. We thank Vitali Fioletov and David Tarasick from Environment and Climate Change Canada (ECCC) for providing Brewer and ozonesonde data. The spring 2006–2014 measurements were also supported by the Canadian Arctic ACE/OSIRIS Validation Campaigns, which were funded by CSA, NSERC, NSTP, ECCC, and the Centre for Global Change Science. We thank CANDAC PI James Drummond, ACE Validation Campaign PI Kaley Walker, PEARL Site Manager Pierre Fogal, the CANDAC operators, and the staff at ECCC's Eureka weather station for their contributions to data acquisition, and logistical and on-site support. We thank Frederick Michel from Carleton University and Xiahong Feng from Dartmouth College for providing Eureka isotopologue precipitation data. This work is also benefitted from discussions with Jonathon Abbatt and Dylan Jones at the University of Toronto. We thank the FLEXPART group for providing the model (https://flexpart.eu/). We thank the ECMWF, GMAO, and NCEP for the meteorological data products. MODIS data were obtained from http://modis.gsfc.nasa.gov/ (false colour images from SwathViewer, http://sv.gina.alaska.edu/).

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





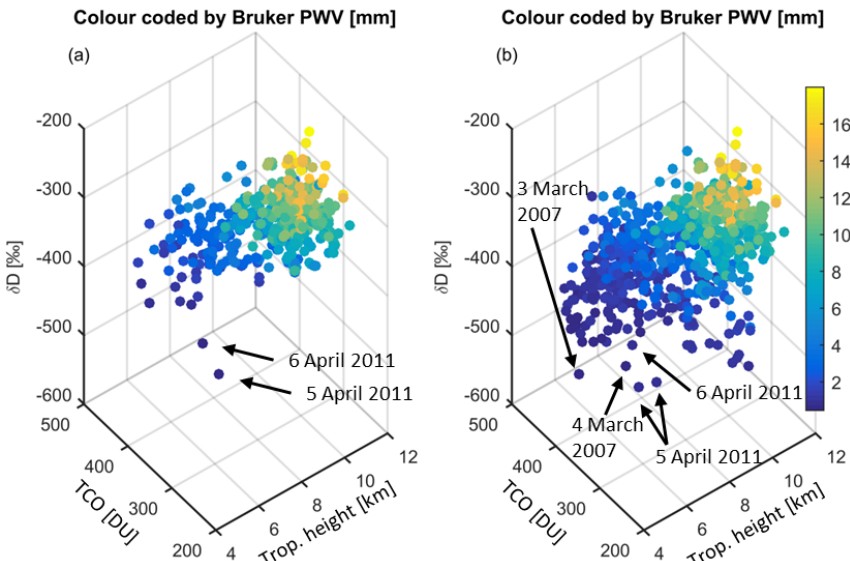

Figure 1: 3-D scatter plots of total ozone column, tropopause height, δD, and total precipitable water. TCO data in (a) are from Brewer no. 69 measurements. TCO data in (b) are from MERRA-2 model output. Tropopause heights are WMO temperature gradient tropopause values calculated from MERRA-2. The δD and PWV are from the Bruker 125HR measurements.





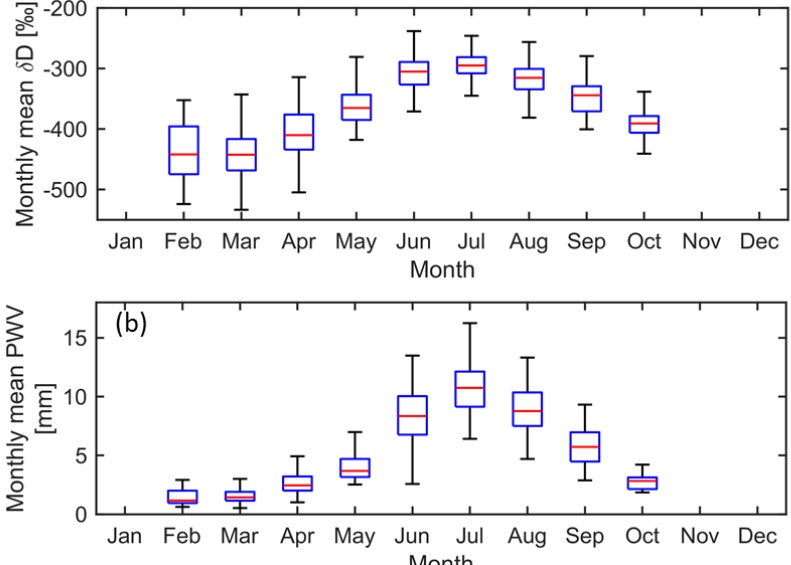

Figure 2: Bruker 125HR monthly (a) δD and (b) PWV. On each box, the central red mark is the median, the edges of the blue box are the 25th and 75th percentiles, and the black whiskers extend to the most extreme data points not considered outliers. The outliers are defined as greater than $q3 + 1.5\ (q3 - q1)$ or less than $q1 - 1.5\ (q3 - q1)$, where $q1$ and $q3$ are the 25th and 75th percentiles of the sample data, respectively (Tukey, 1977).




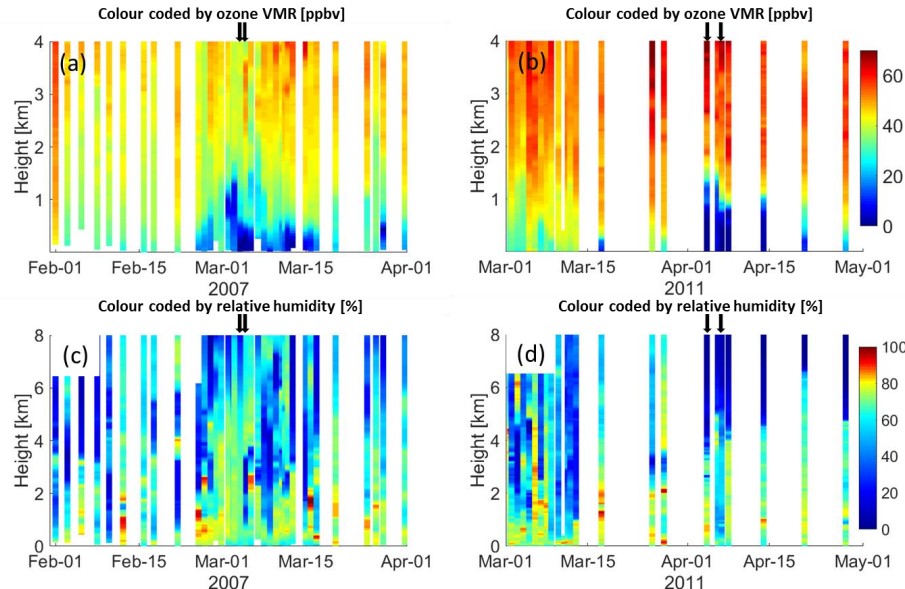

Figure 3: (a) and (b) 0-4 km ozone and (c) and (d) 0-8 km relative humidity profiles above Eureka from ozonesondes and radiosondes. (a) and (c) from 1 February to 1 April 2007; (b) and (d) from 1 March to 1 May 2011. The measurements on 3 and 4 March 2007 and 4 and 6 April 2011 are indicated by the arrows.





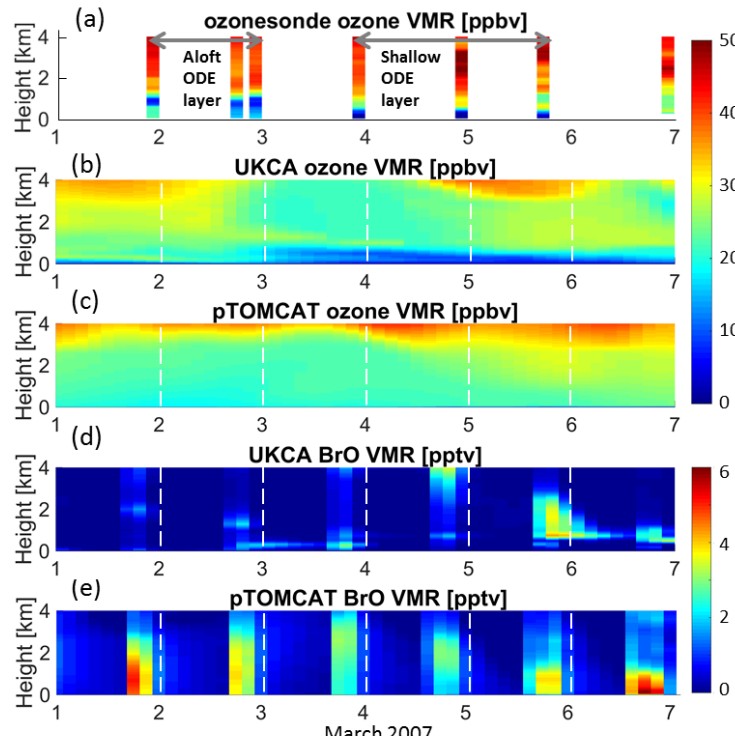

Figure 4: Ozone and BrO volume mixing ratio profiles from 0 to 4 km over Eureka from 1 to 7 March 2007 (UTC): (a) ozonesonde measurements, (b) UKCA modelled ozone profile, (c) pTOMCAT modelled ozone profile, (d) UKCA modelled BrO profile, (e) pTOMCAT modelled BrO profile. Grey horizontal double-headed arrows on (a) indicate the periods when ozonesondes measured the aloft ozone depletion layer and the shallow depletion layer.





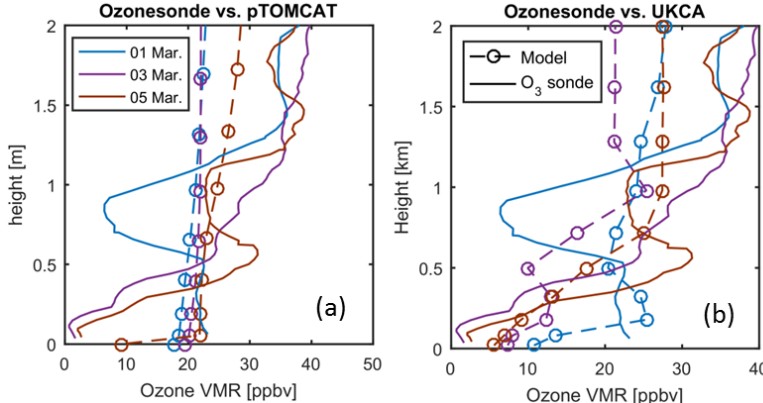

Figure 5: Observed and modelled ozone profiles comparison from 0-2 km on 1, 3, and 5 March: ozonesonde vs. (a) pTOMCAT, (b) UKCA. Modelled profiles are shown by dashed lines with marker o, and measured profiles are shown by solid lines.




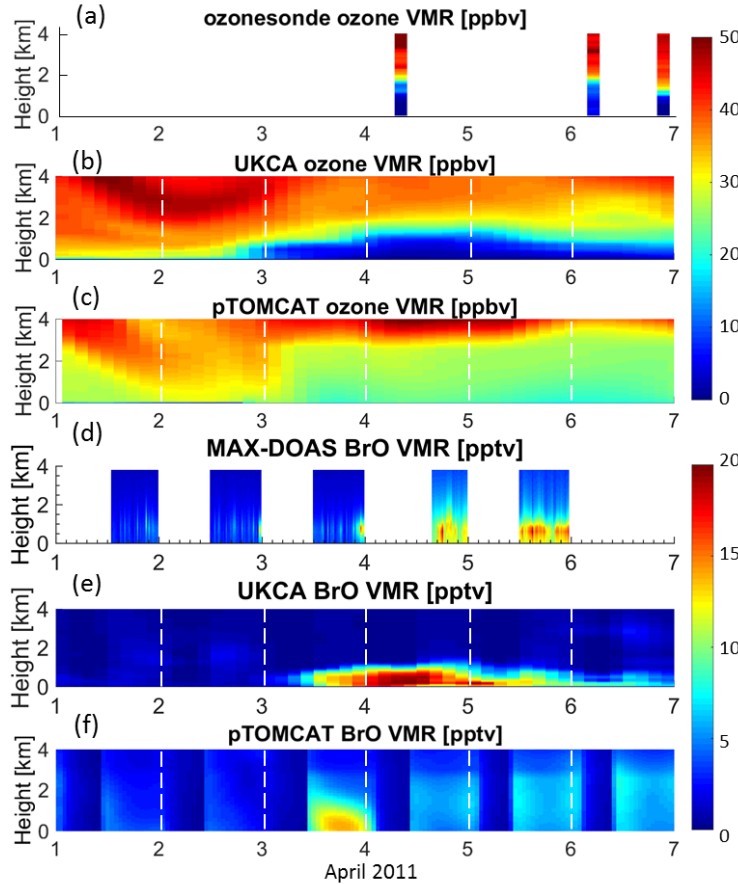

Figure 6: Ozone and BrO volume mixing ratio profiles from 0 to 4 km over Eureka from 1 to 7 April 2011 (UTC): (a) ozonesonde measurements, (b) UKCA modelled ozone profile, (c) pTOMCAT modelled ozone profile, (d) MAX-DOAS retrieved BrO profile, (e) 5 UKCA modelled BrO profile, (f) pTOMCAT modelled BrO profile.





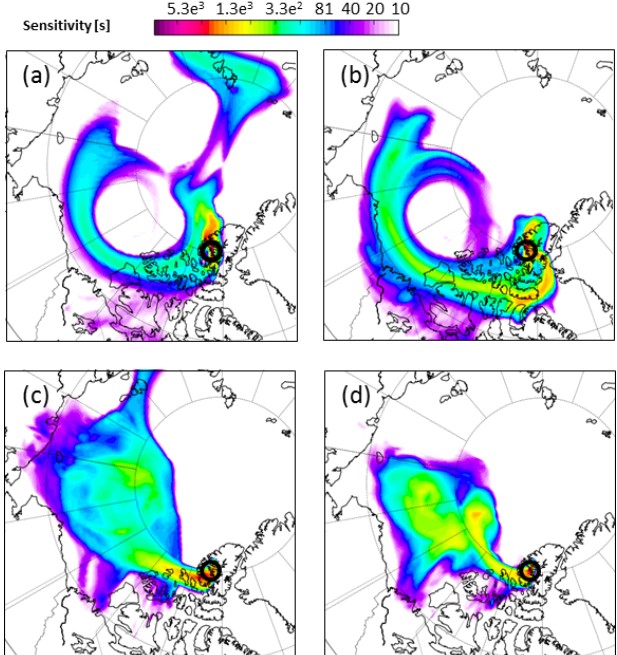

Figure 7: FLEXPART SSA-TRACER 6-day backward run showing total column sensitivity for release from pTOMCAT grid, release times and heights as follows: (a) 17:00-18:00 UTC 3 March 2007, 0-0.5 km, (b) 17:00-18:00 UTC 3 March 2007, 1.5-2.0 km, (c) 14:00-15:00 UTC 3 April 2011, 0-0.5 km, and (d) 14:00-15:00 UTC 3 April 2011, 1.5-2.0 km.





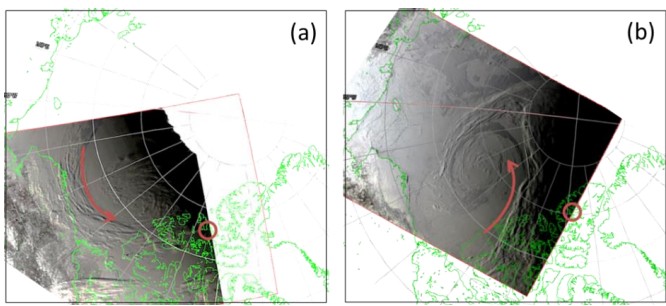

Figure 8: MODIS images show the cyclones (indicated with arrows) over the Beaufort Sea for the 2007 and 2011 events: (a) 27
February 2007, (b) 1 April 2011. Eureka is indicated by the red circle on both panels.



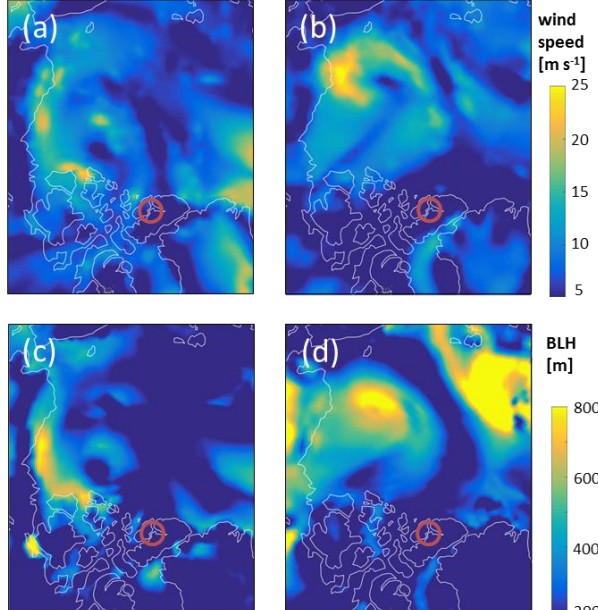

Figure 9: ERA-Interim 10-metre wind gust and boundary layer heights (BLH): (a) wind gust on 28 February 2007 00:00 UTC, (b) wind gust on 1 April 2011 00:00 UTC, (c) boundary layer height at the same time as (a), and (d) boundary layer height at the same time as (b). Eureka is indicated by the red circle on each panel. These two dates were selected when the cyclones were fully developed.



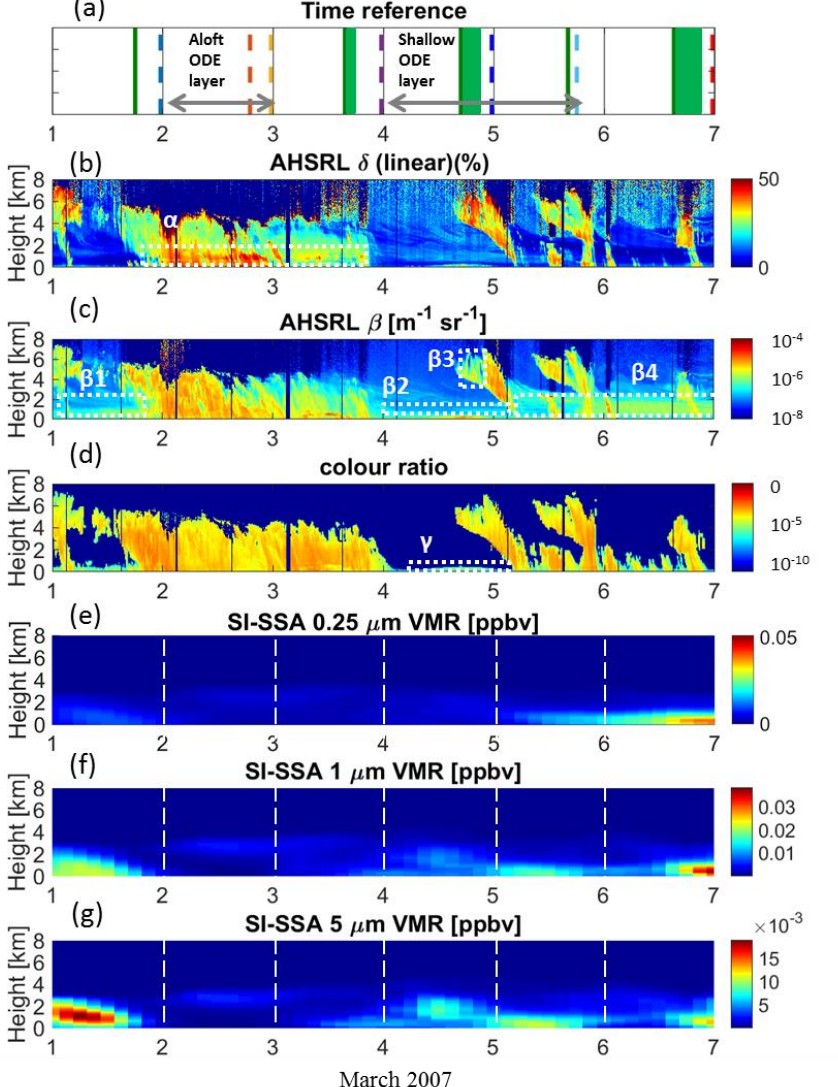

Figure 10: Comparison of lidar (AHSRL) aerosol measurements with pTOMCAT modelled sea-ice sea-salt aerosol (SI-SSA), 0-4 km over Eureka from 1 to 7 March 2007: (a) time reference for coincident measurements from ozonesondes (indicated by colour dashed lines) and Bruker 125HR (indicated by solid green lines/boxes), (b) lidar linear depolarization, (c) lidar backscatter cross-section, (d) colour ratio, (e) pTOMCAT 0.25 μm SISS volume mixing ratio, (f) pTOMCAT 1.0 μm SISS VMR, (g) pTOMCAT 5.0 μm SISS VMR. Grey horizontal double-headed arrows on (a) indicate the periods when ozonesondes measured the aloft ozone depletion layer and the shallow depletion layer. White boxes on (b), (c), and (d) indicate detection of the ice cloud (α), aerosol (β1, β2, and β4), ice cloud (β3), and ice crystal (γ).





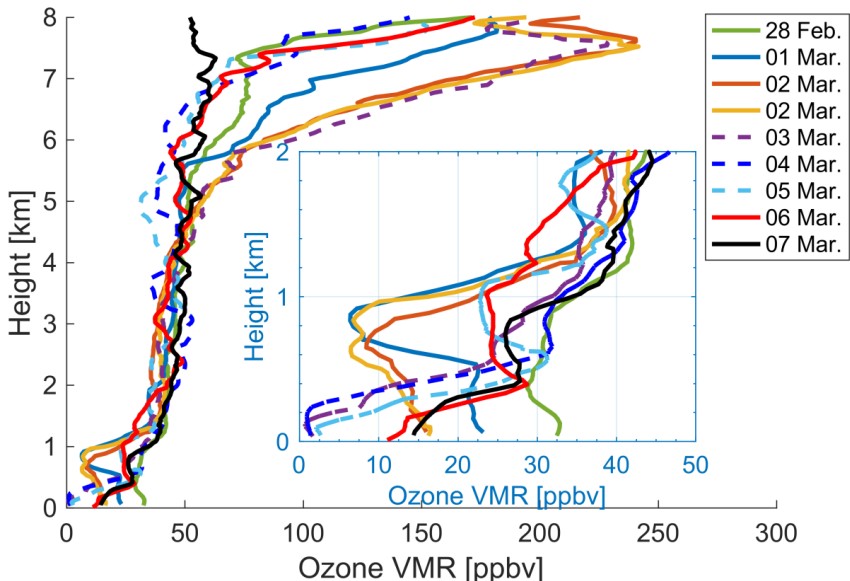

Figure 11: Ozonesonde profiles above Eureka (the inset panel shows 0-2 km) from 28 February to 7 March 2007.



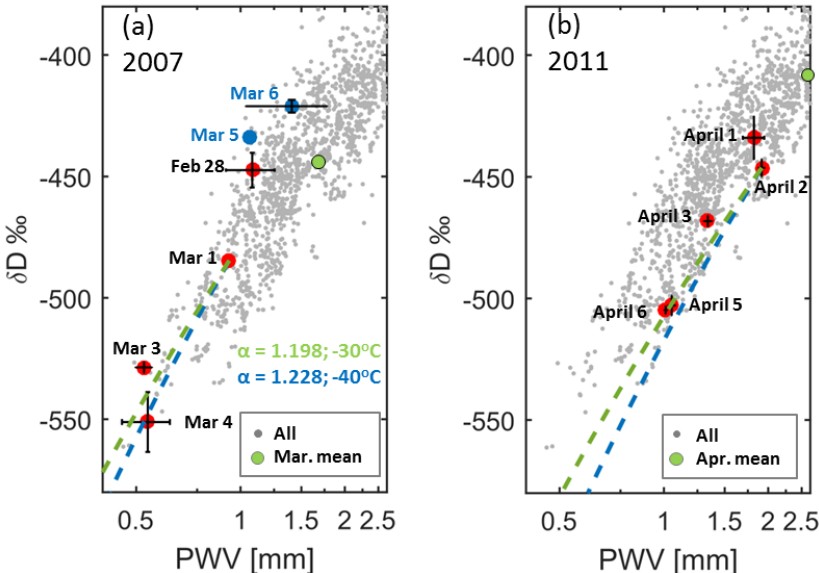

Figure 12: Evolution of δD as a function of PWV (log scale). Grey dots are nine-year (2006-2014) Bruker 125HR δD-PWV measurements. Panel (a) shown 2007 event, with 28 February to 7 March 2007 daily mean δD-PWV values shown in red (a depletion process; decrease of both δD and PWV with time) and blue (a remoistening process; increase of both δD and PWV with time) dots. Panel (b) shown 2011 event, with 1 to 6 April 2011 daily mean δD-PWV values shown by red dots. The green dot in (a)/(b) represents March/April mean values of all nine years. Rayleigh curves with different fraction coefficients are indicated by coloured dashed lines, with (a) using δD-PWV daily mean value on 1 March 2007 as the origin, and (b) using δD-PWV daily mean value on 2 April 2011 value as the origin. The error bars are standard deviations of δD-PWV values used to calculate the daily mean.





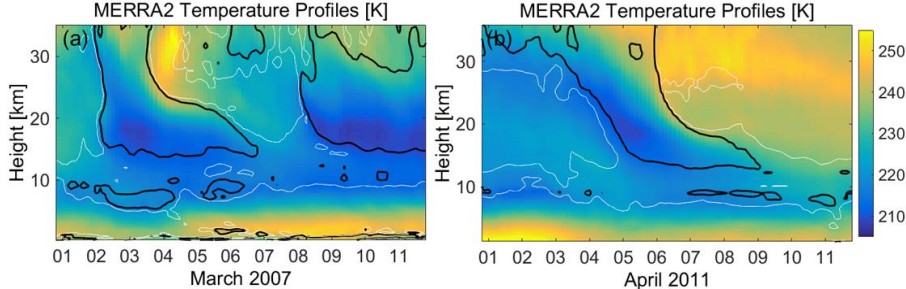

Figure 13: Vertical temperature profiles over Eureka from MERRA-2. The black contour indicates the inner boundaries of the polar vortex determined by sPV = $1.6 \times 10{-4}$ s$^{-1}$, and the white contour indicates the outer boundaries (sPV = $1.4 \times 10{-4}$ s$^{-1}$).





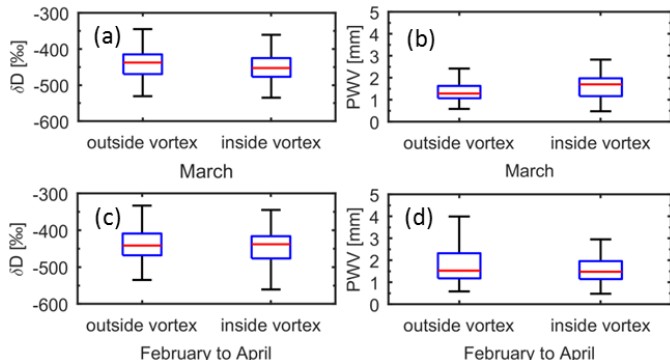

Figure 14: Bruker 125HR δD and PWV whisker plots for nine years of measurements (2006-2014) outside and inside the vortex. (a) and (b) show March δD and PWV measurements respectively, and (c) and (d) show springtime measurements (from February to April). In each box, the central red mark is the median, the edges of the blue box are the 25th and 75th percentiles, and the black whiskers extend to the most extreme data points not considered outliers.