# Peer review of "Cyclone-Induced Surface Ozone and HDO Depletion in the Arctic"

_Atmospheric Chemistry and Physics, 2017_

## Referee Comment (RC1) · Anonymous Referee #1 · 31 Jul 2017

Xiaoyi Zhao et al., focus on two cyclone-induced surface ozone depletion events (ODE) at Eureka, Canada and make connection between ODE and HDO depletions by using ground-based, satellite, and reanalysis datasets. They explain the formation of bromine-enriched SSA from blowing snow processes. In addition, they compare modelling results with their measurements and use FLEXPART to find the potential emission sources. In general, this study is relevant for ACPD/ACP and can help to improve our understanding of ODE process in the Arctic. Making a better connection between SSA and ODE in the introduction could be useful.

Aerosol aloft is necessary, but not sufficient for BrO to distribute vertically (see Simpson et al 2017, ACP). The authors explained the ODE and Br reactions, but it can be interesting if they can add more information about Ozone depletion and the chlorine

radical, too (Custard et al. 2016, ES&T; Custard et al. 2017, ACS Earth & Space Chem).

Equation (2): Are ðİŚŚðİŻ£ changes in $\delta$D and ðİŚŚðİŚđ changes in water vapour mixing ratio?

Page 8 line 9: What is the linear relationship between each pair of the three variables? Can you mention that?

For fig. 1, showing the height in the vertical axis is better.

In fig 1a: what are $\delta$D values for 3rd and 4th of March?

Explain fig 5 more: Is it for 2007? Can you add a plot for 2011 too? Also, why do the model and measurement data disagree in fig 4? Have you measured snow salinity to report here?

Page 2 line 29: ice-covered Page 13 line 10: extremely

---

## Referee Comment (RC2) · Anonymous Referee #2 · 1 Sep 2017

This is the review for the manuscript "Cyclone-induced surface ozone and hydrogen deuterium oxudy (HDO) depletion in the Arctic" submitted by Xiaoyi Zhao et al. The low ozone events have been observed in the Arctic since 1970s (i.e. NOAA Barrow ozone record, Alaska) and were linked to so-called "bromine explosion" events. There are ongoing discussions about sources and mechanisms that are behind the ozone depletion events. Authors analyzed two special ozone depletion episodes observed over Eureka, Canada, which were found to be coincident with HDO depleted-airmasses and the bromine-enriched particles transported with the blowing snow. Observations were compared with chemistry climate model (UKCA) and chemistry transport model (pTOMCAT). The new surface snow salinity scheme was used in both models, which is argued to control the level of the ozone depletion event and amount of transported

[Figure]

BrO. Result of UKCA simulations matched well with observed ozone and BrO levels. Analyses of meteorological conditions associated with the observed low ozone events suggest that the blowing snow creates the aerosols enriched in bromine. The models seem to have limitations with effectiveness of the boundary layer parameterization schemes and the amount of the mixing of the free tropospheric air into the boundary layer. This is well written paper that combines a wide range of measurements, modeled data and reanalyses to investigate the processes in the Arctic. There are several comments and question that should help reader to navigate the discussion of model simulations. Questions and comments: p. 8 line 30, "through", r is missing p. 8, line 37, May be it will be more clear to say "Ozone-sonde observations (Figure 4, a) indicate presence of thin ozone depleted layer at 800~900 m during March 1 and 2 launches that is above the boundary layer". The depleted layer in March 1 profile (Figure 5) seems show either higher boundary layer than in case of two other profiles ( it is hard to tell from ozone profile) or it can be showing growth of the boundary layer that is entrained the tropospheric ozone. Is it possible to discern the height of the boundary later from radiosonde readings? As you pointed out, neither model was capable reproducing the thin (500 m) ozone-depleted layer. The UKCA seems to have a more advanced boundary layer parameterization scheme, but it created a strong boundary condition that is different from ozone-sonde profile. During March 1 and 2 event, the pTOMCAT ozone mixing ratio below 2 km is lower than the UKCA results. Also, pTOMCAT produces large BrO amounts (that also show diurnal cycle) as compared to the UKCA. However, no clear diurnal gradient is observed in the pTOMCAT ozone fields. Also, in UKCA BrO panel, there seems to be a very clear thin shallow boundary layer, with almost no BrO present in the boundary. At the same time, pTOMCAT produces high BrO events that are well mixed all the way to the surface. Since there is a difference in how these two models simulate the boundary layer, there will be differences in their ability to produce ozone depletion events that can be related to the processes in the boundary layer. p.9, lines 1-21. Similarly, the lack of the representative boundary layer mixing scheme in pTOMCAT model deters it from capturing the BrO increase and

ozone depletion in 2011 case, as seen in Figure 6. The issue with the boundary layer dynamics has been mentioned by Authors on p.9, line13-15. I also agree with Authors that the difference between 2007 and 2011 events should be related to the low sunlight conditions. However, it may be also related to the enhanced mixing vs well established boundary layer in 2011. It will be nice to include figure with ozone-sonde and matched model profiles for 2011 case (similar to Figure 5). Then one can understand how stable the boundary layer might have been in 2011 vs 2007. It may be nice to have the Figure that shows time development of the boundary layer over Eureka during studied ozone depletion events. p. 9, lines 32-33, Please explain the units of sensitivity. Does red color mean 100 % sensitivity? p.9 lines 35-39. Discussion of Figure 9. It appears that the boundary layer from ERA-Interim reanalyses over Eureka (circles) on February 28, 2007 stayed very shallow (lower than 200m). Can it be confirmed by comparisons with ozone-sondes? Also, could the reanalyses data for boundary layer be compared with radiosonds/ozone-sondes over Eureka? What was the boundary layer height shown in EAR-Interim data over Eureka on March 1st and March 5th? It can help to understand mixing of the free tropospheric airmasses into the boundary layer. Was the depleted ozone layer observed above the boundary layer on March 1-2 was mixed into boundary on March 3d? I believe that Figure 10, c ( lidar backscatter ratio) can be used for this discussion, showing the top of the enhanced scattering levels to reduce with altitude from ∼ 5 km on March 1 to very shallow layer on March 3, and well mixed boundary on March 4th. It should be possible to analyze the AHSRL data for identification of the boundary layer height? p. 13, lines 8-10 and the previous discussion of the polar vortex influence. The polar vortex mixing into transported airmasses could have happened not over Eureka, but some place further North. Is it possible to do the RDF (reverse domain filling) analyses by using the MERRA 2 or modeled ozone fields to investigate the origin of the airmasses transported over Eureka and how much the ozone in the airmass might have changed during the transport?

---

## Author Comment (AC1) · 6 Oct 2017

**To Referee #1:**

*Xiaoyi Zhao et al., focus on two cyclone-induced surface ozone depletion events (ODE) at Eureka, Canada and make connection between ODE and HDO depletions by using ground-based, satellite, and reanalysis datasets. They explain the formation of bromine-enriched SSA from blowing snow processes. In addition, they compare modelling results with their measurements and use FLEXPART to find the potential emission sources. In general, this study is relevant for ACPD/ACP and can help to improve our understanding of ODE process in the Arctic. Making a better connection*

[Figure]

*between SSA and ODE in the introduction could be useful.*

Thank you to referee #1 for your helpful comments. We have revised the manuscript based on your suggestions. Please note the page and line numbers in our response are referring to the numbers in the "changes tracked" revised manuscript (but the referee's comments refer to numbers in the ACPD version).

*Aerosol aloft is necessary, but not sufficient for BrO to distribute vertically (see Simpson et al 2017, ACP). The authors explained the ODE and Br reactions, but it can be interesting if they can add more information about Ozone depletion and the chlorine*

Thanks for pointing out this relevant work. We include a reference to Simpson et al. (2017) to explain this issue (see p.11, lines 11-16). We agree that the aerosol layer aloft is necessary but not sufficient for bromine aloft. In short, the events (ozone depletion aloft) we reported in this work were due to a different dynamical process (cyclone) compared to the event observed by Simpson et al. (2017) (convection due to the opening of a large sea ice lead). Thus, to distinguish the origin of these different aloft events, multiple measurements (lidar, radar, sondes, etc.) are necessary.

Regarding information about ODEs and chlorine, unfortunately, we do not have any chlorine measurements during the two events studied. In the future, chlorine measurements (such as HOCl and ClO) should be included to facilitate the studies of ODE and mercury deposition.

*Equation (2): Are ???? changes in $\delta D$ and ???? changes in water vapour mixing ratio?*

The symbols in the referee's review are not displayed properly. We guess the question

was: Are $d\delta$ changes in $\delta D$ and $dq$ changes in water vapour mixing ratio?

Yes. We have included this description in the paper (see p5, lines 2-3).

*Page 8 line 9: What is the linear relationship between each pair of the three variables? Can you mention that?*

As pointed out in the manuscript, TCO is not only governed by tropopause height, but also by other factors (e.g., chemical loss). However, in general, there is a simple inverse relation between those three variables. For example, when tropopause height reaches a maximum in summer, TCO is at its minimum value, while $\delta D$ is at its maximum value. In the winter, as the tropopause height decreases, TCO reaches its maximum value, while $\delta D$ is at its minimum value. This general comment has been included in the manuscript (see p.8 lines 2-5).

*For fig. 1, showing the height in the vertical axis is better.*

Fig. 1 has been modified as suggested.

*In fig 1a: what are $\delta D$ values for 3rd and 4th of March?*

For 3 and 4 March 2007, the $\delta D$ values were $-529$‰ and $-551$‰. These numbers were included on p.8 line 14.

In Fig. 1a, there are no data points for the 3rd or 4th of March 2007. This is because the Brewer did not have any TCO measurements in early March, and so there are no paired data points (TCO, h, $\delta D$) for those two days. Thus in Fig. 1b, Brewer TCO was replaced with MERRA-2 TCO, to obtain paired data points for the 3rd and 4th of March

as displayed. This was explained in the manuscript (p.8 lines 11-15).

*Explain fig 5 more: Is it for 2007? Can you add a plot for 2011 too?*

Fig 5 was for the March 2007 event. We have added ozone profile plots for the 2011 event in the revised Fig. 5.

*Also, why do the model and measurement data disagree in fig 4?*

The UKCA model did capture the 2007 shallow surface ODE, as also shown in Fig. 5. However, the aloft ozone depletion layer was not simulated by the model. This could arise from few factors, such as dry/wet deposition velocity of aerosol or model vertical dynamic transport. This disagreement between the model and measurements indicates that improvement are needed for the UKCA model.

On the other hand, pTOMCAT performed worse than UKCA. This is also true for the 2011 event, and is probably due to the models' dynamics. As discussed in Sections 2.2.3 and 3.1.1, UKCA has a more advanced boundary layer dynamics compared to pTOMCAT. Future comparisons between these models are needed to evaluate and improve their performance, but this is beyond the scope of this study.

*Have you measured snow salinity to report here?*

No, we don't have measurements of snow salinity. The snow salinity used in this work for both UKCA and pTOMCAT was the latest surface snow salinity data obtained for the Weddell Sea. The snow salinity used in the model was described in the manuscript (p.7, lines 19-23).

*Page 2 line 29: ice-covered*

Corrected.

*Page 13 line 10: extremely*

Corrected.

**Supplement:**

[revised manuscript text omitted]

---

## Author Comment (AC2) · 6 Oct 2017

**To Referee #2:**

*This is the review for the manuscript "Cyclone-induced surface ozone and hydrogen deuterium oxudy (HDO) depletion in the Arctic" submitted by Xiaoyi Zhao et al. The low ozone events have been observed in the Arctic since 1970s (i.e. NOAA Barrow ozone record, Alaska) and were linked to so-called "bromine explosion" events. There are ongoing discussions about sources and mechanisms that are behind the ozone depletion events. Authors analyzed two special ozone depletion episodes observed over Eureka, Canada, which were found to be coincident with HDO depleted-airmasses*

[Figure]

*and the bromine-enriched particles transported with the blowing snow. Observations were compared with chemistry climate model (UKCA) and chemistry transport model (pTOMCAT). The new surface snow salinity scheme was used in both models, which is argued to control the level of the ozone depletion event and amount of transported BrO. Result of UKCA simulations matched well with observed ozone and BrO levels. Analyses of meteorological conditions associated with the observed low ozone events suggest that the blowing snow creates the aerosols enriched in bromine. The models seem to have limitations with effectiveness of the boundary layer parameterization schemes and the amount of the mixing of the free tropospheric air into the boundary layer. This is well written paper that combines a wide range of measurements, modeled data and reanalyses to investigate the processes in the Arctic. There are several comments and question that should help reader to navigate the discussion of model simulations.*

We appreciate the remarks and suggestions from referee #2, and we have revised the manuscript based on your suggestions. Please note the page and line numbers in our response are referring to the numbers in the "changes tracked" revised manuscript (but the referee's comments refer to numbers in the ACPD version).

*Questions and comments: p. 8 line 30,"through", r is missing*

Corrected.

*p. 8, line 37, May be it will be more clear to say "Ozone-sonde observations (Figure 4, a) indicate presence of thin ozone depleted layer at 800-900 m during March 1 and 2 launches that is above the boundary layer". The depleted layer in March 1 profile (Figure 5) seems show either higher boundary layer than in case of two other profiles*

[Figure]

*( it is hard to tell from ozone profile) or it can be showing growth of the boundary layer that is entrained the tropospheric ozone. Is it possible to discern the height of the boundary later from radiosonde readings? As you pointed out, neither model was capable reproducing the thin (500 m) ozone-depleted layer. The UKCA seems to have a more advanced boundary layer parameterization scheme, but it created a strong boundary condition that is different from ozone-sonde profile. During March 1 and 2 event, the pTOMCAT ozone mixing ratio below 2 km is lower than the UKCA results. Also, pTOMCAT produces large BrO amounts (that also show diurnal cycle) as compared to the UKCA. However, no clear diurnal gradient is observed in the pTOMCAT ozone fields. Also, in UKCA BrO panel, there seems to be a very clear thin shallow boundary layer, with almost no BrO present in the boundary. At the same time, pTOMCAT produces high BrO events that are well mixed all the way to the surface. Since there is a difference in how these two models simulate the boundary layer, there will be differences in their ability to produce ozone depletion events that can be related to the processes in the boundary layer.*

The description of ozonesonde observations for Fig. 4a has been modified as requested (see p.8 line 30 to p.9 line 2).

We also calculated the boundary layer heights (BLH) from radiosondes as suggested. We found that the thermal BLH on 1 March 2007 was 960 m, whereas the thermal BLH on 3rd March decreased to 440 m. The potential temperature profiles from radiosondes (see the new Fig. 5c) indicate the typical strong inversion layer in the Arctic spring was not present on 1 March. Further discussion of the BLH (Fig. 5) can be found in the later part of this response.

We agree with the referee that boundary layer parametrization played important role in the ozone depletion simulations. As discussed in the manuscript, UKCA and pTOMCAT used different boundary layer parametrizations. Comparing the simulations with measurements, this work reveals that improved modelling of boundary layer

dynamics is necessary to further improve ODE simulations. Also, the discussion of atmospheric boundary layer stability has been rephrased and added to the revised paper (p.9, lines 2-5).

*p.9, lines 1-21. Similarly, the lack of the representative boundary layer mixing scheme in pTOMCAT model deters it from capturing the BrO increase and ozone depletion in 2011 case, as seen in Figure 6. The issue with the boundary layer dynamics has been mentioned by Authors on p.9, lines 13-15. I also agree with Authors that the difference between 2007 and 2011 events should be related to the low sunlight conditions. However, it may be also related to the enhanced mixing vs well established boundary layer in 2011. It will be nice to include figure with ozonesonde and matched model profiles for 2011 case (similar to Figure 5). Then one can understand how stable the boundary layer might have been in 2011 vs 2007. It may be nice to have the Figure that shows time development of the boundary layer over Eureka during studied ozone depletion events.*

We have modified Fig. 5 to include ozone profiles for the 2011 case, and also radiosonde thermal boundary layer height information (and ERA-Interim BLH). In fact, we found both events to have high thermal boundary layers (> 900 m) compared to normal conditions in Eureka (< 500 m). We agree with the referee that, besides sunlight conditions, the difference between the 2007 and 2011 events could be related to the difference in boundary layer mixing. For example, the potential temperature profile for 4 April 2011 shows a clearer decrease trend from the surface up to about 1 km, which indicates stronger boundary layer mixing compared to 1 March 2007. A discussion of boundary layer heights has been included (p.9, lines 2-7).

*p. 9, lines 32-33, Please explain the units of sensitivity. Does red color mean $100\%$ sensitivity?*

Fig. 7 is the FLEXPART "retroplume" output, which shows emission sensitivity in unit of second. When multiplied with emission fluxes, the sensitivity yields a simulated concentration at the receptor (Stohl et al., ACP 2013 ). This information was included, and more details (unit and release flux) have been added (p.9, lines 31-32).

*p. 9 lines 35-39. Discussion of Figure 9. It appears that the boundary layer from ERA-Interim reanalyses over Eureka (circles) on February 28, 2007 stayed very shallow (lower than 200m). Can it be confirmed by comparisons with ozone-sondes? Also, could the reanalyses data for boundary layer be compared with radiosonds/ozone-sondes over Eureka? What was the boundary layer height shown in EAR-Interim data over Eureka on March 1st and March 5th? It can help to understand mixing of the free tropospheric airmasses into the boundary layer. Was the depleted ozone layer observed above the boundary layer on March 1-2 was mixed into boundary on March 3d? I believe that Figure 10, c ( lidar backscatter ratio) can be used for this discussion, showing the top of the enhanced scattering levels to reduce with altitude from $\sim 5km$ on March 1 to very shallow layer on March 3, and well mixed boundary on March 4th. It should be possible to analyze the AHSRL data for identification of the boundary layer height?*

On Feb 28, 2007, the ERA-Interim BLH was 161 m over Eureka (not shown in the paper) and the radiosonde thermal BLH was 126 m.

We have added two panels (c and f) in Fig. 5 to illustrate the boundary layer mixing conditions for these two events. Figures 5c and f show that the thermal boundary layer heights (calculated from radiosonde measurements) on 1 March 2007 and 4 April 2011 were both about 1 km, which is much higher than the boundary layer heights on 3 to 5 March 2007 and 6 April 2011. The potential temperature profiles indicate the breakdown of the surface temperature inversion when the cyclones arrived. Compared to the radiosonde thermal boundary layer heights, ERA-Interim boundary layer data indicate that the surface mixing conditions for the 2007 event and the 2011 event are very different. The ERA-Interim BLH is based on eddy diffusion combined with mass-flux transport (see Dee et al., The ERA-Interim reanalysis: configuration and performance of the data assimilation system, Q. J. R. Meteorol. Soc., 2011). It is possible that the ERA-interim BLH data underestimated mixing strength for the 2007 event given the potential temperature profile on 1 March 2007.

The AHSRL provides a vertical profile of the backscatter cross-section. In boundary layers with strong vertical mixing, a sharp gradient in the backscatter cross-section at the top of the layer can be often be used to mark the the boundary layer depth. However, in not well mixed atmospheres (such as observed at Eureka for these two events), the vertical distribution of aerosol may be controlled by processes that occurred far away and then transported to the site, or by changes in the local relative humidity causing particle sizes to change as the air cools or warms. Thus identifications of BLH from AHSRL data for these two events are difficult.

*p. 13, lines 8-10 and the previous discussion of the polar vortex influence. The polar vortex mixing into transported airmasses could have happened not over Eureka, but some place further North. Is it possible to do the RDF (reverse domain filling) analyses by using the MERRA 2 or modeled ozone fields to investigate the origin of the airmasses transported over Eureka and how much the ozone in the airmass might have changed during the transport?*

We agree that the suggested simulations would be interesting, but they are beyond the scope of this work. The study of the polar vortex influence (Section 3.3.2) was to illustrate that the $\delta D$ variation during those two events was caused by tropospheric HDO depletion. This work helps confirm that the HDO depletion reported in the paper

provides evidence for blowing-snow sublimation processes in the troposphere.

As addressed in the paper, if the polar vortex mixing into the transported airmass happened before it arrived at Eureka, the radiosonde relative humidity (RH) profiles should reveal layers that have low RH and high ozone concentration. However, this typical stratosphere-troposphere exchange (STE) signature was not found in any of the radiosonde measurements. We also checked the ozone field from MERRA-2 over Eureka, and no clear evidence of STE (e.g., elevated ozone values) was found.

[Figure]

**Supplement:**

[revised manuscript text omitted]